# Separation and Quantification of Selected Sapogenins Extracted from Nettle, White Dead-Nettle, Common Soapwort and Washnut

**DOI:** 10.3390/molecules26247705

**Published:** 2021-12-20

**Authors:** Magdalena Ligor, Anna Kiełbasa, Ileana-Andreea Ratiu, Bogusław Buszewski

**Affiliations:** 1Department of Environmental Chemistry and Bioanalytics, Faculty of Chemistry, Nicolaus Copernicus University, 7 Gagarina Str., 87-100 Torun, Poland; kielbasam@umk.pl; 2“Raluca Ripan” Institute for Research in Chemistry, Babes-Bolyai University, 30 Fantanele Str., RO-400239 Cluj-Napoca, Romania; andreea_ratiu84@yahoo.com; 3Interdisciplinary Centre of Modern Technologies, Nicolaus Copernicus University, 4 Wileńska Str., 87-100 Torun, Poland

**Keywords:** plant material, sapogenins, extraction liquid chromatography, antioxidant activity

## Abstract

Saponins are an important group of secondary metabolites naturally occurring in plants with important properties like: antibacterial, antiviral and antifungal. Moreover, they are widely used in the cosmetic industry and household chemistry. The sapogenins are saponin hydrolyses products, frequently used to facilitate saponin detection. In the present study, an improved methodology for isolation and separation of five sapogenins extracted from nettle (*Urtica dioica* L.), white dead-nettle (*Lamium album* L.), common soapwort (*Saponaria officinalis* L.) and washnut (*Sapindus mukorossi* Gaertn.) was developed using ultra-high-performance liquid chromatography with an evaporative light-scattering detector (UHPLC-ELSD). Based on quantitative analysis, the highest content of hederagenin (999.1 ± 6.3 µg/g) and oleanolic acid (386.5 ± 27.7 µg/g) was found in washnut extracts. Good recoveries (71% ± 6 up to 99% ± 8) were achieved for four investigated targets, while just 22.2% ± 0.5 was obtained for the fifth one. Moreover, hederagenin and oleanolic acid of whose highest amount was detected in washnut (999.1 ± 6.3 µg/g and 386.5 ± 27.7 µg/g, respectively) were subject to another approach. Consequently, liquid chromatography coupled mass spectrometry (LC/MS) with multiple reaction monitoring mode (MRM) was used as an additional technique for fast and simultaneous identification of the mentioned targets.

## 1. Introduction

Plant materials, due to the bioactive compounds content, including polyphenols, terpenes, saponins, cyclitols and others, are used in various areas of life. They are food for humans and animals; they have the ability to cure, but some of them can even kill living organisms [1,2]. Some plants, especially herbs, due to the high content of biologically active compounds are widely used in various treatments [3,4]. Nettle (*Urtica dioica* L.) and white dead-nettle (*Lamium album* L.) are well known as plants with antioxidant properties [5,6,7]. Due to the presence of polyphenols, tannins, sterols, fatty acids and polysaccharides, these plants also possess antibacterial, analgesic, anti-inflammatory, antiviral and anticancer activities [5,6,7]. The abovementioned plants are widely used in traditional medicine and besides this, they have other practical uses, for example, as ingredients in shampoos and shower gels.

The sapogenins are saponin hydrolyses products, frequently used to facilitate saponin detection. Saponins are an important group of secondary metabolites naturally occurring in plants. They are detected in samples like: common soapwort (*Saponaria officinalis* L.) or washnut (*Sapindus mukorossi* Gaertn.) [8,9,10,11]. These plants are widely used in the cosmetic industry and as household chemicals, e.g., washing powder. The saponins are incompletely soluble in two different solvents, a hydrophilic and a hydrophobic one. As can be observed in Figure 1, saponins are constituted by the aglycone sapogenin and the glycone saccharide. In the sugarless fragment, from 4 to 6 cyclical carbon rings may be found, while the sugar part contains from 1 to 3 branching chains consisting of 1 to 6 monosaccharide particles. In saponin structure, the most abundant sugars are glucose, galactose, rhamnose, arabinose, xylose and fructose. Glucuronic and galacturonic acid can be found, but they are more rare. In the general structure, the aglycone is connected to the glycone by an ether bond or ester bond, while the hydroxyl groups of sugars are acetylated multiple times. The molecular mass of saponins usually ranges from 600 to 1500 amu [12].

Identification and quantification of saponins is feasible by using a large spectrum of analytical techniques. The most relevant to be mentioned are: thin-layer chromatography (TLC), high-performance liquid chromatography (HPLC) or ultra-high-performance liquid chromatography (UHPLC) with various detectors (mass spectrometer (MS), UV-visible, diode array detector (DAD) or evaporative light-scattering detector (ELSD)) and gas chromatography with: mass spectrometer (GC-MS) or with flame ionization detector (GC-FID) and column chromatography [2]. A good option to detect saponins is the identification of their hydrolyzed products, which are the sapogenins. The obtaining of sapogenins can be achieved by a hydrolysis process either under mild or strong conditions, using different temperature gradients and by involving hydrochloric acid or sulfuric acid, in various concentrations. The extracts can be prepared in water, methanol or dioxane [13,14]. The drawback of this approach is the artifact occurrence; however, purification methods are available for their removal [15]. TLC as a standalone technique can be used just for separation, while another technique must be used for detection and quantification. Separation of saponins using TLC was realized using a mixture of n-butanol:water:acetic acid (84:14:7 *v*/*v*/*v*) [16]. The analysis of saponins by GC-MS or GC-FID, even if possible, is probably not the best option. The main shortcoming is the too-low volatility of the saponins for being identified by GC-MS. While the identification of sugar units could be possible from their trimethylsilyl derivatives, when more than one type of sugar is present, multiple isomeric sugar forms detection is another issue [17]. However, GC-MS was used for identification of sapogenins, while the quantification was achieved by their methyltrimethylsilyl derivatives using a GC-FID system. Nineteen sapogenins were detected this way, from six different morphological parts of *Medicago arabica* L. [18]. According to the ample evidence from articles in the field, HPLC with various detectors is the most used technique for identification and quantification of saponins or their derivatives. HPLC-UV combined with the usage of a pre-column derivatization when a chromophoric system (4-bromophenacyl bromide) is introduced was used for saponins containing free carboxyl groups [14]. In turn, HPLC-MS made possible the determination of saponins without their derivatization [19,20,21]. Both UPLC/Q-TOF MS and HPLC-UV were used for identification and quantification of ginsenosides extracted from *Panax notoginseng* leaves. A total of 57 saponins were detected from the extracts. The authors concluded that a possible enzymatic hydrolysis in the presence of water was responsible for the transformation of ginsenosides, the main components of *Panax notoginseng*, in notoginsenosides [22]. Reverse-phase HPLC was used for the determination of two hydrolyzed saponins: hederagenin and oleanolic acid in *Flos Lonicerae Japonicae*. The hydrolysis step was realized with sulfuric acid and extracted with chloroform. The authors reported very good average recoveries (more than 98%) for both targets. Both enzymatic and acid hydrolysis were involved in the isolation of four sapogenins (medicagenic acid, bayogenin, hederagenin and soyasapogenol B), while UHPLC-ELSD was used for quantification. High-quality recoveries (71 to 99%) were obtained when C18 was used for purification, while just 29 to 45% was reported in the case of purification using OASIS^®^ MCX [15]. The qualitative and quantitative analyses of saponins in samples by use of LC/MS are required in various fields from scientific research to manufacturing. Various analytical solutions are proposed in this field. In exemplary selected ion monitoring (SIM), the analysis is conducted by use of MS sets to scan over a very small mass range, typically one mass unit. Only compounds with the selected mass are detected [23]. Besides the SIM method, multiple reaction monitoring (MRM) is widely applied. MRM is the method which involves the performance of mass spectrometric quantitation. The MRM mode allows the analysis to be accomplished by specifying the parent mass of the compound for MS/MS fragmentation and then specifically monitoring for a single fragment ion.

In general, the knowledge about saponins and physicochemical properties of this large group of compounds, while also taking into account their biological activity, can distinguish many possibilities for their practical use. The proposals and various possibilities of saponin use obtained from natural sources, resulting mainly from their physicochemical properties and biological activity, are presented in Figure 2.

The aim of this work was to carry out qualitative and quantitative analysis for determination of five selected sapogenins extracted from nettle, white dead-nettle, common soapwort and washnut. The novelty of the present study consists in proposing an improved methodology of fast simultaneous determination quantification of five sapogenins by using simple and environmentally friendly techniques. A short time for sample preparation and simplicity of methods handling characterized the whole experiment. A very small amount of solvents (~hundreds of microliters) was required in all steps of analysis, while the longer time needed for the hydrolysis process was just two hours. The SPE column recovery was higher than 80% for three of the five investigated targets. Moreover, for hederagenin and oleanolic, extracted from washnut, targets obtained in high concentration compared with the others, a fast LC-MS identification method based on MRM was presented and discussed. Extracts from the investigated plants can represent valuable sources for the cosmetic industry as well as for manufacturing of domestic detergents.

## 2. Results and Discussion

### 2.1. Antioxidant Activity of Investigated Plants

The antioxidant activity of extracts, from nettle, white dead-nettle, common soapwort and washnut with the concentration 0.04 g of dry material/mL was studied. Inhibition times for the studied extracts were 15 min. After those periods, absorbance was measured (λ = 517 nm). The results concerning radical scavenging activity appearing as *RSA* are presented in Table 1. Total antioxidant activity of the plant extracts was calculated using the following Formula (1):(1)$$RSA=\frac{A_{DPPH}-A}{A_{DPPH}}\cdot 100\%$$
where: *A*—absorbance of the mixture of DPPH^●^ with plant extract after inhibition time; *A_DPPH_*—absorbance of pure DPPH^●^ solution; *RSA*—radical scavenging activity.

The proper inhibition time of the reaction between synthetic free radical DPPH^●^ and compounds from extracts influences the higher results of evaluated antioxidant activity. Nevertheless, the highest results for radical scavenging activity were observed for washnut and common soapwort, while for nettle and white dead-nettle the results were lower.

### 2.2. Separation of Plant Extract Components Using TLC Analysis

Obtained plant extracts containing various types of biologically active compounds including saponins were subjected to acid hydrolysis as a required method to obtain unbound sapogenins. First of all, to examine the composition and characteristics, preliminary chromatographic analyses of obtained extracts after acid hydrolysis were performed by use of TLC. The proposed mobile phase was a mixture of toluene:methanol 90:10 (*v*/*v*), rather than a non-polar mobile phase applicable to sapogenins [12]. Extracts from nettle leaves are characterized by a significantly different qualitative composition when compared to other plant samples, in particular, white dead-nettle (flowers), common soapwort (roots) and washnut. Obtained chromatograms are also presented in Table 2. Observations made by us allow us to indicate that the richest in terms of the composition were extracts from nettle leaves. Due to the presence of polyphenols and chlorophylls in nettle extracts, using chromatograms detected at wavelength λ = 366 nm, the largest number of components was observed. However, the use of diphenylboric acid 2-aminoethyl ester (NP) for the derivatization reaction also confirmed the presence of flavonoids in other extracts, which was taken into account.

### 2.3. Qualitative Analysis by Means of LC/MS of the Most Abundant Targets

Washnut was the richest source in investigated components; however, just two targets, oleanolic acid and hederagenin with concentration of 386.5 ± 27.7 and 999.1 ± 6.3, respectively, were detected. Consequently, washnut extract was the subject of LC/MS analysis, in an attempt to investigate another approach to sapogenin identification. MRM mode was used. Some conditions of LC/MS analysis of standard solutions including mobile phase, flow rate and injection volume have been optimized. Obtained results made it possible to identify hederagenin (t_R_ = 32.004 ± 0.100 min) and oleanolic acid (t_R_ = 37.255 ± 0.130 min). In Figure 3A,B the MS spectra of the hederagenin and oleanolic acid standards with the application of MRM mode are presented, while in Figure 3C an example of LC/MS chromatogram of washnut extract is presented. The proposed MRM delivers a unique fragment pattern of ion from analyte that can be monitored and quantified in complex matrix [23]. For this reason, MRM mode can be suitable for a sensitive and specific quantitation as well. In the present work, MRM mode is accomplished by specifying the parent mass of the compound selected for fragmentation and then specifically monitoring for a single fragment of ion. Based on the obtained *m*/*z* values of the deprotonated ions (*m*/*z* 471.1 and 455.2), the fragmentation product ions could be observed.

To evaluate the reliability of the applied procedure, each water solution obtained from tested plants was spiked with hederagenin and oleanolic acid solutions in methanol (concentration level 0.100 µg/mL). As has been indicated, acceptable recoveries were obtained in the range 85–105% for tested samples.

### 2.4. Quantitative Analysis by Use of UHPLC/ELSD

From the analytical point of view, an applied method is usually validated in terms of linearity, precision, repeatability, as well as accuracy [24,25]. For this reason, the quantitative analysis was performed by an external standard method, were known data from a calibration standard and unknown data from the sample are combined to generate a quantitative report. From the basic standard solution, several working standard solutions were prepared by successive dilutions. For each of the compounds, basic validation parameters and the detection and determination limits were calculated (Table 3). LOD and LOQ were calculated based on signal to noise ratio. The analytes were eluted in the following order: medicagenic acid (t_R_ = 20.54 min), bayogenin (t_R_ = 21.82 min), hederagenin (t_R_ = 24.88 min), soyasapogenol B (t_R_ = 29.69 min) and oleanolic acid (t_R_ = 31.92 min).

Calibration curves were prepared in different concentration ranges for each standard such as: medicagenic acid (0.006–0.37 mg/mL), bayogenin (0.004–0.26 mg/mL), hederagenin (0.005–0.34 mg/mL), soyasapogenol B (0.005–0.34 mg/mL) and oleanolic acid (0.009–0.30 mg/mL), respectively. Correlation coefficient was in the range from 0.9978 to 0.9990. Moreover, the following were evaluated: the limit of detection (LOD) from 0.01 to 0.05 μg/mL and the limit of quantification (LOQ) from 0.03 to 1.5 μg/mL. Purification of the hydrolyzed extracts was necessary. The recovery for SPE with octadecyl sorbent for sapogenins was between 22 and 99%, as presented in Table 3.

However, the results of quantitative analysis regarding methanol extracts of nettle, white dead-nettle, common soapwort and washnut are presented in Table 4 and graphically in Figure 4. It was ascertained that all sapogenins, medicagenic acid, bayogenin, hederagenin, soyasapogenol B and oleanolic acid, are present in extracts of common soapwort. Extracts from washnut are characterized by the highest concentration of hederagenin (999.1 ± 6.3 μg/g). Methanolic extracts of nettle and white dead-nettle do not contain sapogenins besides oleanolic acid in low concentration, 81.0 ± 0.7 and 76.0 ± 2.7 μg/g, respectively. It was ascertained that sapogenins such as medicagenic acid, bayogenin and soyasapogenol were not present in any extract of washnut.

### 2.5. Surface Tension of Washnut Solution Containing Saponins

Due to the high concentration of the determined sapogenins in the tested material, especially for washnut, further studies were carried out confirming the possibility of their practical application. Water solutions of washnut at various concentrations were tested. The obtained data allowed us to determine the surface tension of the tested solutions by use of Formula (2) mentioned below:σ_x_ = σ_0_ (h_x_/h_water_) [mN m^−1^](2)
where: σ_0_ = 71.97 [mN/m]—surface tension of water; h_x_—difference in levels in the pressure gauge for tested solution [mm]; h_water_—operating pressure gauge level for distilled water (120 mm).

The results obtained for carried out solutions depending on the concentration change are presented in Table 5 and in Figure 4. The first changes in surface tension were observed for concentrations ranging from 400 to 600 ppm. The reduction in the surface tension to ca. 48–42 mN/m with the use of solutions in a concentration range from 800 to 1000 ppm indicates the possibility of wide application of washnut as a detergent. As shown in Figure 5, the surface tension of aqueous solutions dropped from 56.98 mN/m to 26.99 mN/m in 1000 ppm washnut water extract.

The test results obtained depending on the mixing time are presented in Table 6 and Figure 6. The surface tension in the first 3 h of mixing did not significantly change (constant value of σ = 44.98 mN/m). A noticeable drop occurred only at 4 and 5 h of mixing. It follows that the mixing time of the solution does not significantly change the surface tension.

Water extracts from the investigated plants can certainly be used in a variety of ways, especially for the cosmetic industry as well as for the manufacturing of domestic detergents. For the large group of compounds like saponins, taking into account their physicochemical properties, and also their biological activity, we can distinguish many possibilities for their practical use. This predisposes saponins obtained from natural sources to a high position when it comes to practical use, resulting mainly from their physicochemical properties and biological activity [26,27,28,29,30,31,32]. As confirmed by numerous studies and which has also been proven in this work, a number of valuable properties of saponins deserve attention. This group of compounds is known as emulsifiers and long-lasting foaming agents [26], combating pests of crops, adversely affecting insects or other animals, as well as being molluscicidal [27]. Moreover, some saponins possess expectorant and bronchodilator properties, reduce cholesterol levels, can be applied for synthesis of steroid drugs and show biological activity, including fungicidal, antibacterial, cytotoxic, antiviral, antitumor and others [28,29,30,31,32]. The highest content of sapogenins occurs in washnut and common soapwort, making it possible for these plants to be used as valuable sources for the production of natural, environmentally friendly cosmetics and domestic detergents. Due to the presence of all investigated compounds in roots of common soapwort, it can be used in cosmetics as well.

## 3. Materials and Methods

### 3.1. Plant Material

Different morphological parts of plants were used for extraction. More specifically: (1) leaves of nettle—Urticae folium (*Urtica dioica* L.), bought from Kawon, Gostyń, Poland; (2) flowers of white dead-nettle—Lamii albi flos (*Lamium album* L.), purchased from FLOS Co., Mokrsko, Poland; (3) roots of common soapwort—Saponariae radix (*Saponaria officinalis* L.) acquired from Herba-Vita, Pińczów, Poland; (4) hulls of washnut (*Sapindus mukorossi* Gaertn.), originating from India, were purchased in a local herbal store from Toruń, Poland. The dry plant material was ground in a mortar with pestle, and after that in a laboratory mill.

### 3.2. Chemicals and Reagents

Standards of sapogenins: medicagenic acid, bayogenin and soyasapogenol B (with purity 97%) were supplied by Trimen Chemicals (Łódź, Poland). Standards of hederagenin (purity 99.35%) were purchased from Roth (Karlsruhe, Germany) and oleanolic acid (purity 97%) was purchased from Sigma Aldrich (St. Louis, MO, USA). The stock solutions were prepared by dissolving sample weight of mentioned compounds in 1 mL methanol and lower concentrations were prepared by diluting the stock solution with methanol. Methanol, acetonitrile, toluene and acetic acid were purchased from J.T. Baker (Deventer, Holland). Hydrochloric acid (36–38%) was purchased from Avantor Performance Materials S.A. (Gliwice, Poland). Synthetic free radical 2,2-diphenyl-1-picrylhydrazyl (DPPH^●^) was purchased from Sigma Aldrich (St. Louis, MO, USA). Water was obtained by means of MilliQ RG apparatus by Millipore Intertech (Bedford, MA, USA). BAKERBOND^®^ Octadecyl (C18, 500 mg) was purchased from J.T. Baker.

### 3.3. Sample Extraction and Purification

Before extraction, 1 g of washnut was weighed and ground and 25 mL of distilled water was added to the sample. Then it was mixed on a magnetic stirrer at 500 rpm for 60 min. The obtained extract was filtered through filter paper with 0.2 μm pore size and subjected to analysis. The same procedure was used to obtain extracts of common soapwort. The purification by SPE using C18 was preceded by column conditioning. At first, methanol was passed through the column (5 mL, twice) followed by passing of 5 mL of distilled water to prevent the extraction column from drying. Then the column was filled with 5 mL of extract at the flow rate 0.2 mL/min. Finally, the adsorbed components were eluted with 3 mL of pure methanol. The same procedure was applied to obtain recovery results. Thus, the column was filled with 5 mL of solution containing standards in the concentration level 0.200 μg/mL.

### 3.4. Antioxidant Activity

The proposed method was partially carried out according to Bajkacz et al. 2021 [33]. Crude methanolic extracts from a nettle, white dead-nettle, common soapwort and washnut were investigated for an evaluation of antioxidant activity. To obtain a solution of DPPH^●^, 2.00 mg of pure substance was dissolved in 100 mL of methanol (concentration 0.02 mg/mL; 50.7 μM). Next, 100 μL of methanolic extracts was added to 3 mL of DPPH solution and kept in darkness for 15 min reaction time. At the end of these periods, absorbance was measured with a spectrophotometer Helios δ (Thermo Fisher Scientific Inc., Waltham, MA, USA) at the wavelength λ = 517 nm. The absorbance of pure DPPH solution was also measured, with pure methanol used for the blind experiment.

### 3.5. Surface Tension Measurements

The surface tension measurements were partially carried out according to Ceynowa et al., 2006 [34]. Water solutions of washnut at the concentration 400, 500, 600, 700, 800, 900 and 1000 ppm were tested. Solutions were prepared in beakers by weighing the appropriate amount of ground washnut shells and mixing with 100 cm^3^ of distilled water at room temperature. Obtained solutions were mixed by use of a magnetic stirrer at a speed of 1000 rpm, temperature during measurements 22.5 °C, mixing time 3600 s. The surface tension of prepared water solutions was measured using the well-known Rebinder’s method; besides a thermostat and pressure gauge, a properly selected capillary was applied. An important role was played by the difference of liquid level value in the manometer measured for distilled water and considered solutions. This was due to the disruption of the surface film caused by the contact between the capillary and the liquid surface and the appearance of air bubbles. Additionally, five weights of washnut shells were prepared and mixed with 100 cm^3^ of distilled water to obtain the final concentration 200 ppm. These solutions were mixed with a magnetic stirrer at 1000 rpm at various time intervals (for 1, 2, 3, 4 and 5 h). In each type of experiment, data were expressed for three independent experiments.

### 3.6. Procedure of Hydrolysis and Extract Purification

Obtained plant extracts were hydrolyzed (acid hydrolysis) and purified (clean-up procedure) by the use of modified methods that were previously described by Bajkacz et al., 2021 [33]. Briefly, 300 µL plant extract was mixed with 200 µL distilled water and 100 µL concentrated hydrochloric acid. The mixture was incubated at 85 °C for 2 h. Next, aqueous hydrolyzed extract was purified using SPE with BAKERBOND^®^ Octadecyl. The analytes were eluted with methanol, and the extract was concentrated in a stream of nitrogen. Through the hydrolysis process, unbound sapogenins were formed by removal of the sugar part from the saponins, according with the reaction presented in Figure 7.

### 3.7. TLC Analysis

Prepared plant extracts were analyzed by the use of the same method, with modification, that was described in our previous work [33]. HPTLC system from CAMAG (Muttenz, Switzerland) including a Linomat V Applicator, Visualizer and vision CATS data processor (version 2.0) was used. TLC analysis of standards and plant extracts was performed in a DS-L horizontal chamber obtained from Chromdes (Lublin, Poland). TLC plates with silica gel on aluminum foil and plastic background Kieselgel 60 F254 (Merck, Darmstadt, Germany) were also used. The plates were observed at UV light (λ = 254 nm and λ = 366 nm). Mobile phase consisting of toluene:methanol 90:10 (*v*/*v*) was prepared. Plates, sized 10 × 10 cm, were covered with methanol extracts of the mentioned plants. The chromatograms were processed at the distance of 9.0 cm. The chromatograms on silica gel plates were processed for 30 min. The plates were dried (at 40 °C) and observed in UV light and were subsequently submerged in the reagent causing diphenylboric acid 2-aminoethyl ester (NP). The reagent used was methanol solution of NP at the concentration of 10 g/L. Detection was performed at the wavelengths λ = 254 and 366 nm. Some obtained chromatograms have been presented in Table 2.

### 3.8. Instrumentation Used for Sample Analysis

Two types of equipment have been used for qualitative and quantitative analysis of standard solutions and plant extracts. The chromatographic analysis was performed with an HPLC instrument consisting of quaternary pump 1100 (Agilent Technologies, Waldbronn, Germany). The detection system consisted of MS Triple Quad 6410 mass spectrometer (Agilent Technologies, Waldbronn, Germany). Moreover, the liquid chromatograph 1260 Infinity Series (Agilent Technologies, Waldbronn, Germany) with an evaporative light-scattering detector 380-ELSDSeries (Agilent Technologies, Waldbronn, Germany) was also used. Required nitrogen gas purity was higher than 99%. The Eppendorf^®^ Thermomixer Comfort was used for sample hydrolysis at controlled conditions. MULTIVAP-8 Automated Concentration Solution (LabTech, Sorisole, Italy) was used for evaporation of solvents from the samples.

### 3.9. Quantitative Analysis by Use of UHPLC/ELSD

The quantitative analysis was partially carried out according to de Combarieu et al., 2002 and Gao et al., 2015 [24,25]. The quantification of targets in obtained extracts and standard solutions was performed by means of UHPLC/ELSD on a column Aqua 5u C18 125A (150 × 4.60 mm; 5 μm), obtained from Phenomenex. Acetonitrile and water were mobile phase and contained glacial acetic acid (0.05%, *v*/*v*). The following gradient program is recommended: 0–5 min; 25% acetonitrile with acid, 5–30 min; 25–90% acetonitrile with acid, 30–35 min; 90–25% acetonitrile with acid, 35–40 min; 25% acetonitrile with acid. The mobile phase was set at the flow rate of 1.0 mL/min. The column temperature was set at 35 °C and 10 μL of the sample was injected. Evaporation temperature of the ELSD was set at 80 °C and nebulizer temperature was set at 40 °C. The gas flow was 2.0 mL/min. The particular calibration data including: calibration equations, linearity coefficient (R^2^), LOD, LOQ and recovery have been presented in Table 3.

### 3.10. Qualitative Analysis by Use of LC/MS

The qualitative analysis was carried out following the method of Chen et al. 2011 [23] with modification. The SPE extracts of washnut, the richest source, were analyzed by means of LC/MS. The analysis conditions are presented in Table 7. The following parameters were used for the analysis: data mode—total ion current (TIC) in mass range 105–1000, fragmentor 150; drying gas temperature 325 °C; nitrogen flow 9 L/min; capillary voltage 4000 V; nebulizer 35 PSI.

MRM mode was used (details: negative polarization, dwell 200, fragmentor 150 V, collision energy 55 V); work mode: full scan from 0 to minute 30; MRM *m*/*z* 471→393 from minute 30 to 34; full scan from minute 34 to 36; MRM *m*/*z* 455→407 from minute 36 to 39; full scan from minute 39 to 42. Peak assignment for oleanolic acid and hederagenin was performed first by mass spectrum library NIST 2005 (Gaithersburg, MD, USA) and confirmed by comparing with the retention times of pure standards. Examples of results have been presented in Figure 3.

## 4. Conclusions

The methodology applied for the separation and determination of five sapogenins (medicagenic acid, bayogenin, hederagenin, soyasapogenol B and oleanolic acid) in four plants has been developed. LC/MS with MRM mode made possible the determination of the *m*/*z* value of deprotonated ions for hederagenin *m/z* = 471.1 and for oleanolic acid *m*/*z* = 455.2. The proposed methodology for quantification of the mentioned components, by UHPLC-ELSD technique, presented high reproducibility. The richest sapogenins content could be found in washnut and common soapwort, which can constitute valuable sapogenin sources for production of cosmetics and domestic detergents. Due to the presence of medicagenic acid, bayogenin, hederagenin, soyasapogenol B and oleanolic acid in roots of common soapwort can thus be used in cosmetics as well. Based on the physicochemical properties and biological activity of saponins, the search for natural sources of these plant-derived compounds, which will also be more environmentally friendly detergents, is a challenge for scientists and should be continued.

## Figures and Tables

**Figure 1 molecules-26-07705-f001:**
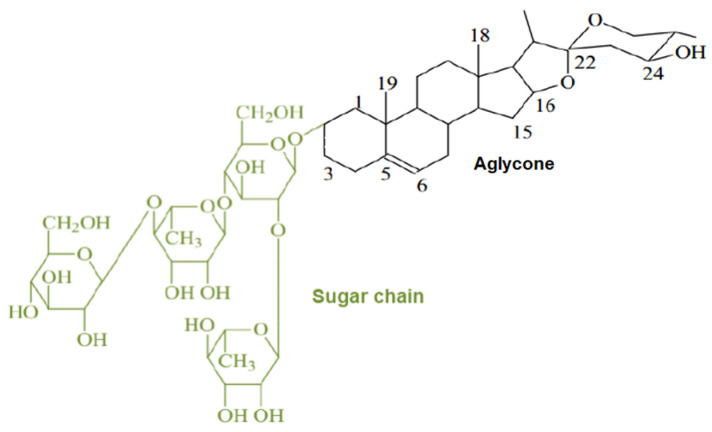
General structure of saponins.

**Figure 2 molecules-26-07705-f002:**
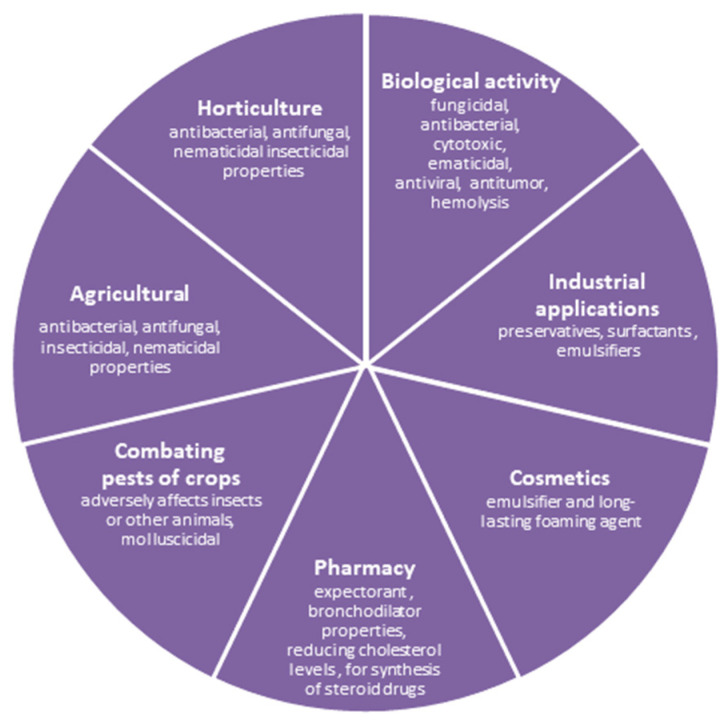
Applications of saponins due to their physicochemical and biological properties as well as their featured structural diversity.

**Figure 3 molecules-26-07705-f003:**
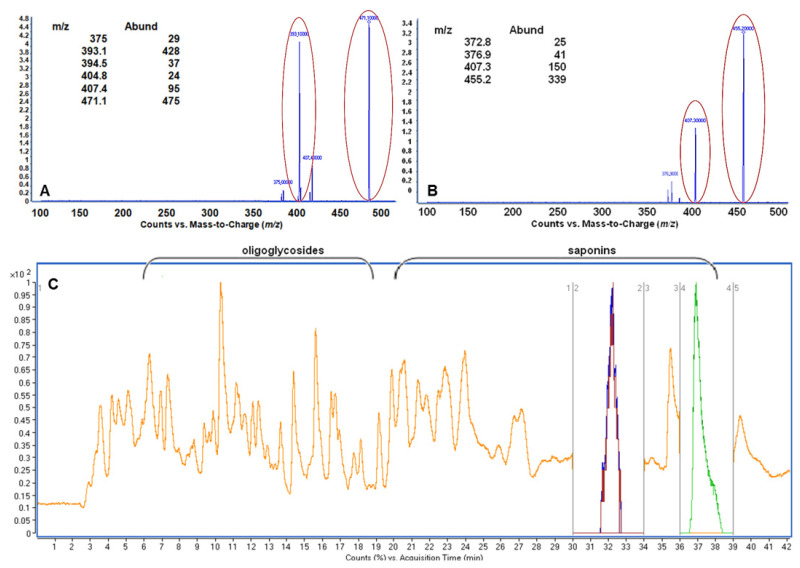
MS spectra of hederagenin (**A**; negative polarization, *m*/*z* 471.1), oleanolic acid (**B**; negative polarization, *m*/*z* 455.2) and example of LC/MS chromatogram of washnut extract (**C**).

**Figure 4 molecules-26-07705-f004:**
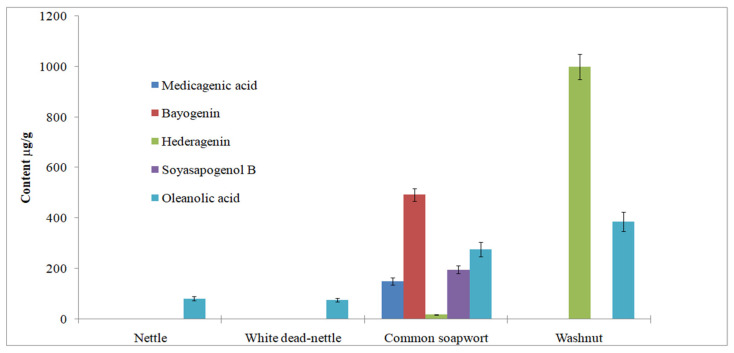
The quantitative content of medicagenic acid, bayogenin, hederagenin, soyasapogenol B and oleanolic acid in extracts of nettle, white dead-nettle, common soapwort and washnut.

**Figure 5 molecules-26-07705-f005:**
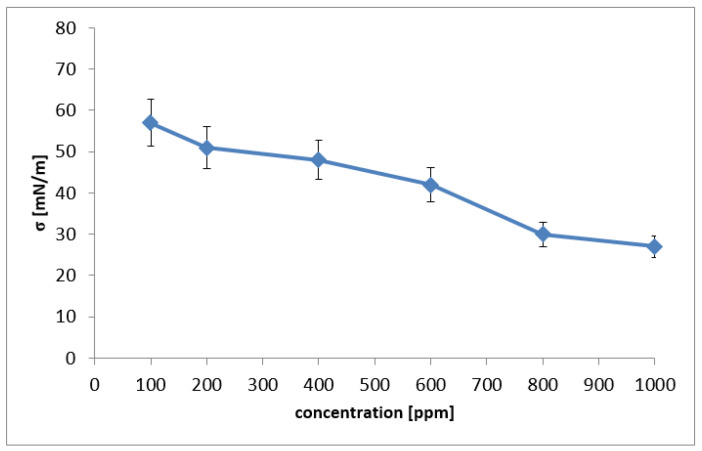
The variation of the surface tension versus the increasing solution concentration. Data are expressed as mean ± SD of three independent experiments. (-♦-) washnut water solution.

**Figure 6 molecules-26-07705-f006:**
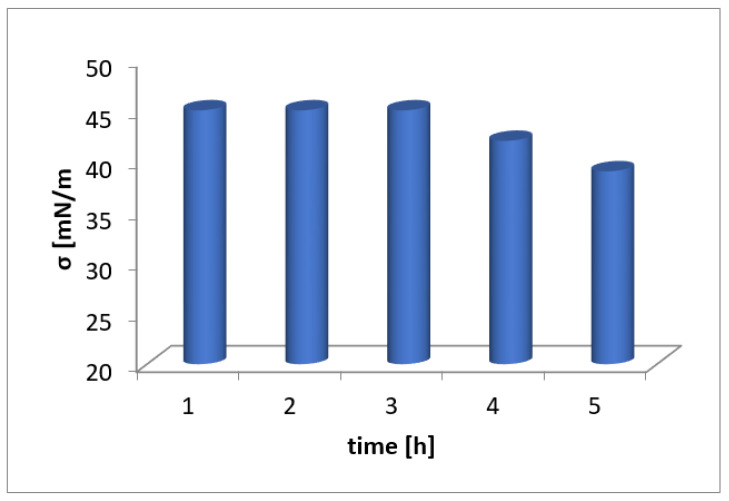
The variation of the surface tension versus mixing time of solutions containing washnut.

**Figure 7 molecules-26-07705-f007:**
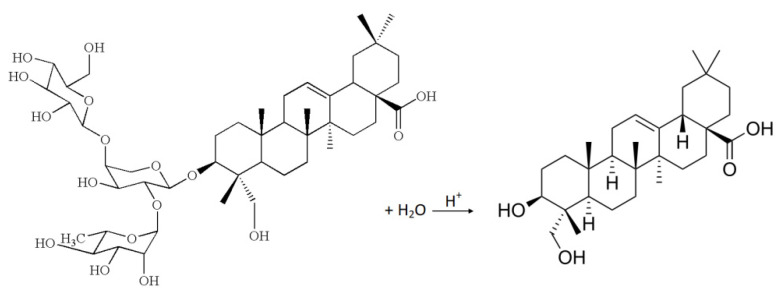
Sapogenin formation during an acid hydrolysis (hederagenin as example).

**Table 1 molecules-26-07705-t001:** Total antioxidant activity after 15 min of an inhibition time.

**Sample**	**Nettle** **(*Urtica dioica* L.)** 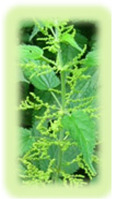	**White Dead-Nettle** **(*Lamium album* L.)** 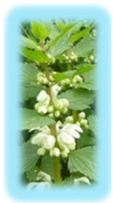	**Common Soapwort** **(*Saponaria officinalis* L.)** 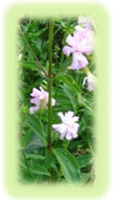	**Washnut** **(*Sapindus mukorossi* Gaertn.)** 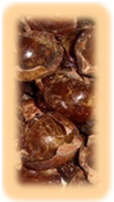
**A****± SD [%]** absorbance at λ = 517 nm	0.389 ± 0.001	0.303 ± 0.001	0.230 ± 0.002	0.197 ± 0.001
**A_DPPH_****± SD [%]** absorbance at λ = 517 nm	0.969 ± 0.002			
**RSA ± SD [%]**	**59.82 ± 0.06**	**68.77 ± 0.12**	**76.26 ± 0.21**	**79.70 ± 0.06**

Where: RSA—radical scavenging activity; SD—standard deviation.

**Table 2 molecules-26-07705-t002:** Comparative analysis of extracts from the studied plants.

	Detection	TLC Chromatogram λ = 254 nm	TLC Chromatogram λ = 366 nm	TLC Chromatogram Derivatized by NP λ = 254 nm	TLC Chromatogram Derivatized by NP λ = 366 nm
Sample No					
**(1) Nettle** **(2) White dead-nettle** **(3) Common soapwort** **(4) Washnut**		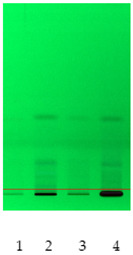	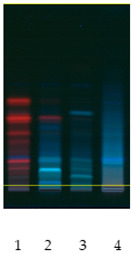	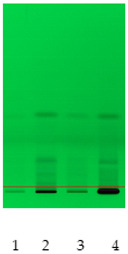	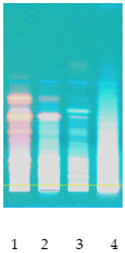

**Table 3 molecules-26-07705-t003:** Calibration data including: calibration equations, linearity coefficient (R^2^), LOD, LOQ and recovery (*n* = 3).

Compounds	Equation of the Calibration Curve	Correlation Coefficient R^2^	LOD [µg/mL]	LOQ [µg/mL]	Recovery ± SD [%]
Medicagenic acid	y = 1.1026x − 0.6054	0.9986	0.50	1.50	71.0 ± 6.0
Bayogenin	y = 1.3236x − 0.9934	0.9978	0.30	0.90	83.1 ± 5.0
Hederagenin	y = 1.4434x − 1.0916	0.9983	0.10	0.30	87.2 ± 1.7
Soyasapogenol B	y = 1.3884x − 1.1674	0.9990	0.20	0.70	99.0 ± 8.0
Oleanolic acid	y = 1.4109x − 1.4243	0.9985	0.30	1.10	22.2 ± 0.5

**Table 4 molecules-26-07705-t004:** Results of quantitative analysis of extracts obtained by HPLC-ELSD technique.

	Extract	Nettle Content ± SD [µg/g]	White Dead-Nettle Content ± SD [µg/g]	Common Soapwort Content ± SD [µg/g]	Washnut Content ± SD [µg/g]
Compound					
Medicagenic acid		nd	nd	149.3 ± 6.3	nd
Bayogenin		nd	nd	492.7 ± 19.8	nd
Hederagenin		nd	nd	17.9 ± 1.6	999.1 ± 6.3
Soyasapogenol B		nd	nd	195.3 ± 16.0	nd
Oleanolic acid		81.0 ± 0.7	76.0 ± 2.7	276.2 ± 35.6	386.5 ± 27.7

**Table 5 molecules-26-07705-t005:** Results of the surface tension versus the increasing solution concentration.

Concentration [ppm]	h_x_ [mm]	σ_x_ [mN/m]
100	95	56.98
200	85	50.98
400	80	47.98
600	70	41.98
800	50	29.99
1000	45	26.99

**Table 6 molecules-26-07705-t006:** Results of surface tension of extracts at different mixing times.

Sample No	Mixing Time [h]	h_x_ [mm]	σ_x_ [mN/m]
1	1	75	44.98
2	2	75	44.98
3	3	75	44.98
4	4	70	41.98
5	5	65	38.98

**Table 7 molecules-26-07705-t007:** Analysis conditions for plant extracts.

Column	C_18_ Column (150 × 4.6 mm; 5.0 μm)		
Mobile phase	A: water/acetic acid (0.01%)		
	B: acetonitrile/acetic acid (0.01%)		
Mobile phase flow rate	0.6 mL/min		
Gradient elution	Time [min]	A [%]	B [%]
	0	75	25
	15	55	45
	25	55	45
	26	20	80
	40	20	80
Injection	10 μL		
UV-Vis detector	λ = 205 nm		

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
