# Peer review of "Separation and Quantification of Selected Sapogenins Extracted from Nettle, White Dead-Nettle, Common Soapwort and Washnut"

_molecules, 2021, doi:10.3390/molecules26247705_

Round 1
Reviewer 1 Report
The manuscript ‘Separation and quantification of selected sapogenins extracted from
nettle, white dead-nettle, common soapwort and washnut’ is well planned, well executed and well written. However, I found some major and minor issues exist with the current form of the manuscript. The method of separation is pretty common and I am little not convinced with the novelty of the work. The manuscript requires major changes and a considerable reorganization with elaborate discussion. I am enlisting few points which the authors may consider incorporating in the revised manuscript.
- Line 36: Please add some other biological properties or applications.
- Line 65: Please modify the sentence “acidor sulphuric acid’ pl give space between words.
- The novelty of the work is not impressive. Please elaborate MRM and explain the innovativeness of the extraction process as LC/MS is an old practice of compound separation and quantification.
- The result section must be supported with discussion and appropriate references. Please discuss the results with references.
- Line 138: Modify sentence “was mixture tolulene” to “was mixture of tolulene”.
- Line 142: Results obtained should be explained elaborately (Table 2).
- Line 151: Modify sentence “The analysis of what ??? made it possible”.
- Line 157-160: Please check the sentence and modify.
- Line 133-142, 146-160: Reference is missing in the whole paragraph.
- Line 173-174: Please re-write the sentence.
- Line 239-240: Pl avoid mentioning reference in a queue.
- Line 363-372: Please conclude your manuscript with more take home messages. Please elaborate the conclusion with significant findings and its future prospects/applications.
- Add reference in material and methods is missing. The methodology followed need to be supported with references.
- Please cite some published papers as reference in the results and discussion section. Add some elaborated discussion/theories vis-à-vis hypothesis in the result section comparing with your observations. It is advisable to use significant measures to distinguish and elaborate the results in the bar (in the writing part).
- References may be cross checked as per the journal pattern.
- The author may revise the manuscript and re-submit.
The manuscript may be revised thoroughly and resubmitted.
Author Response
The reply to the Referee
I would like to thank for the thorough and examination of our paper. I tried to revise our manuscript in accordance with reviewers' comments. I hope that the revised work will satisfy reviewers.
Responding to the Review 1
- Line 36: Please add some other biological properties or applications.
Answer: It has been completed. Due to present of important phytochemicals such as polyphenols, tanins, sterols, fatty acids, polysaccharides mentioned plants exhibit also many biological activities, among others antibacterial, analgesic, anti-inflammatory, antiviral, anticancer [5-7].
- Line 65: Please modify the sentence “acidor sulphuric acid’ pl give space between words.
Answer: It has been corrected.
- The novelty of the work is not impressive. Please elaborate MRM and explain the innovativeness of the extraction process as LC/MS is an old practice of compound separation and quantification.
Answer: The qualitative and quantitative analysis of saponins in samples by use of LC/MS are re-quired in various fields from scientific research to manufacturing. Various analytical so-lutions are proposed in this field. Exemplary selected ion monitoring (SIM) the analysis by use of MS sets to scan over a very small mass range, typically one mass unit. Only com-pounds with the selected mass are detected [23]. Beside SIM method multiple reaction monitoring (MRM) is widely applied. MRM is the method used by the performing of mass spectrometric quantitation. The MRM mode allows to accomplished by specifying the parent mass of the compound for MS/MS fragmentation and then specifically monitoring for a single fragment ion.
- The result section must be supported with discussion and appropriate references. Please discuss the results with references.
Answer: Whenever literature in this field was available, we tried to use all data.
Water extracts from the investigated plants can certainly be used in a variety of ways, especially for the cosmetic industry as well as for the manufacturing of domestic detergents. For the large group of compounds like saponins, taking into account their physicochemical properties, and also their biological activity, we can distinguish many possibilities of their practical use. This predisposes high position of saponins obtained from natural sources to practical use, resulting mainly from their physicochemical properties and biological activity. The available scientific reports confirm this thesis [26-32].
- Line 138: Modify sentence “was mixture tolulene” to “was mixture of tolulene”.
Answer: It has been corrected.
- Line 142: Results obtained should be explained elaborately (Table 2).
Answer: It has been completed. Observations made by us allow to indicate that the richest in terms of the composition were extracts from nettle leaves. Due to the presence of polyphenols and chlorophylls in nettle extracts, on chromatograms detected at wavelength λ = 366 nm the largest number of components was observed. However, the use of diphenylboric acid 2-aminoethyl ester (NP) for the derivatization reaction also confirmed the presence of flavonoids in other extracts were taken into account.
- Line 151: Modify sentence “The analysis of what ??? made it possible”.
Answer: It has been corrected. Obtained results made it possible to identify hederagenin (tR = 32.004 ± 0.100 min) and oleanolic acid (tR= 37.255 ± 0.130 min).
- Line 157-160: Please check the sentence and modify.
Answer: It has been corrected. For this reason MRM mode can be suitable for a sensitive and specific quantitation as well. In the present work, MRM mode accomplished by specifying the parent mass of the compound selected for fragmentation and then specifically monitoring for a single fragment of ion.
- Line 133-142, 146-160: Reference is missing in the whole paragraph.
Answer: It has been completed. Reference [12] and [23] were taken into consideration.
- Line 173-174: Please re-write the sentence.
Answer: For thais reason the quantitative analysis was performed by an external standard method, were known data from a calibration standard and unknown data from the sample are combined to generate a quantitative report.
- Line 239-240: Pl avoid mentioning reference in a queue.
Answer: The surface tension of prepared water solutions was measured using a well-known Rebinder's method, besides a thermostat and pressure gauge, a properly selected capillary was applied.
- Line 363-372: Please conclude your manuscript with more take home messages.
Answer: The methodology applied for the separation and determination of five sapogenins (medi-cagenic acid, bayogenin, hederagenin, soyasapogenol B and oleanolic acid) in four plants has been developed. LC/MS with MRM mode made possible the determination of the m/z value of deprotonated ions for hederagenin m/z= 471.1 and for oleanolic acid m/z= 455.2. The proposed methodology for quantification of the mentioned components, by UHPLC-ELSD technique, presented high reproducibility. The richest sapogenins content could be found in washnut and common soapwort, which can constitute valuable sapo-genin sources for production of cosmetics and domestic detergents. Due to presence of medicagenic acid, bayogenin, hederagenin, soyasapogenol B and oleanolic acid in rots of common soapwort this can be used in cosmetics as well. The search for natural sources of plant-derived saponins, which will be more environmentally friendly compounds, is a challenge for scientists and should be continued.
- Please elaborate the conclusion with significant findings and its future prospects/applications.
Answer: The highest content of sapogenins in washnut and common soapwort, makes it possible to use of these plants as constitute valuable sources for the production of natural, environmentally friendly cosmetics and domestic detergents. Due to presence of all investigated compounds in rots of common soapwort, it can be used in cosmetics as well.
- Add reference in material and methods is missing. The methodology followed need to be supported with references.
Answer: Our experiments based on home-made methods as well as references (Chen et al., 2011; de Combarieu et al., 2002; Gao et al., 2015). We have taken it into account in the new version of the manuscript.
- Please cite some published papers as reference in the results and discussion section. Add some elaborated discussion/theories vis-à-vis hypothesis in the result section comparing with your observations. It is advisable to use significant measures to distinguish and elaborate the results in the bar (in the writing part).
Answer: Some references including study of Chen et al., 2011; de Combarieu et al., 2002; Gao et al., 2015; Huang et al., 2003; Oleszek and Marson, 2000. Sparg et al., 2004; Takagi et al. 1980; Tamura et al., 2001; Tanaka et al., 1996 were taken into consideration.
- References may be cross checked as per the journal pattern.
Answer: References have been checked and established as recommended.

Reviewer 2 Report
dear authors,
plz mention figure 6 in introduction.
some of the tables in results (the details of the method) need to be mentioned in materials and methods.
Author Response
The reply to the Referee
I would like to thank for the thorough and examination of our paper. I tried to revise our manuscript in accordance with reviewers' comments. I hope that the revised work will satisfy reviewers.
Responding to the Review 2
Comment 1
plz mention figure 6 in introduction.
Answer: Suggestion was taken into account. The Figure was moved to the Introduction part.
Comment 2
some of the tables in results (the details of the method) need to be mentioned in materials and methods.
Answer: The information has been completed. Chapter Materials and methods has been completed.

Reviewer 3 Report
Separation and quantification of selected sapogenins extracted from nettle, shite-nettle…
This work presents the qualitative and quantitative analysis in order to determine five selected sapogenins extracted from nettle, white dead-nettle, common soapwort and washnut. The novelty consists of proposing an improved methodology by using simple and environmentally friendly techniques. Whashnut was the richest source in investigated components, however just two targets, oleanoic acid and hederagenin with appreciable concentrations were detected.
The description of the experimental part is clear and complete. The techniques used were LC / MS with MRM mode.
The results were validated in terms of linearity, precision, repeatability, and accuracy. The surface tension of water solution of washnut at various concentrations were determined confirming the possibility of their practical application.
References are adequate and up dated
I do recommend publishing this work in Molecules
Author Response
The reply to the Referee
I would like to thank for the thorough and examination of our paper. I tried to revise our manuscript in accordance with reviewers' comments. I hope that the revised work will satisfy reviewers.
Responding to the Review 3
Answer: We would like to thank Reviewer for the positive assessment of our research. The study in this research area is continued and results are still estimated.
Reviewer 4 Report
The authors did a good work from an experimental point of view and I recommend the article for publication after some minor revisions.
More specific:
L282: Report the concentration of DPPH solution in units of concentration (e.g., μM).
L283: kept in the darkness for 15. Minutes? Mention it. Usually, in most methods the inhibition time was 30 minutes.
L284: Model and company of spectrophotometer are missing.
Author Response
The reply to the Referee
I would like to thank for the thorough and examination of our paper. I tried to revise our manuscript in accordance with reviewers' comments. I hope that the revised work will satisfy reviewers.
Responding to the Review 4
L282: Report the concentration of DPPH solution in units of concentration (e.g., μM).
Answer: DPPH solution was prepared at the concentration 0.02 mg/mL (50.7 μM)
L283: kept in the darkness for 15. Minutes? Mention it. Usually, in most methods the inhibition time was 30 minutes.
Answer: According to our previous investigations, it was confirmed that the free radical DPPH● reacted most intensively with the antioxidant tested up at time period 3 - 15 minutes and after that the reaction was often reversible. It is also significant that in this method the concentration of the test substance was important for the course of the reaction velocity. Given the above, we decided to applie 15 minutes reaction time.
L284: Model and company of spectrophotometer are missing.
Answer: spectrophotometer Helios δ (Thermo Fisher Scientific Inc., Waltham, MA, USA)

Round 2
Reviewer 1 Report
The manuscript ‘Separation and quantification of selected sapogenins extracted from
nettle, white dead-nettle, common soapwort and washnut’ has been substantially improved to the level of acceptance. I am enlisting a few points which the authors may address prior to acceptance.
- Please follow the ‘Instruction to authors @ Molecules’, The heading ‘Discussion’ is missing. The authors may mention ‘Results and Discussion’ in section 2 or make ‘Result’ and ‘Discussion’ separately.
[Research Manuscript Sections
- Introduction: The introduction should briefly place the study in a broad context and highlight why it is important. It should define the purpose of the work and its significance, including specific hypotheses being tested. The current state of the research field should be reviewed carefully and key publications cited. Please highlight controversial and diverging hypotheses when necessary. Finally, briefly mention the main aim of the work and highlight the main conclusions. Keep the introduction comprehensible to scientists working outside the topic of the paper.
- Results: Provide a concise and precise description of the experimental results, their interpretation as well as the experimental conclusions that can be drawn.
- Discussion: Authors should discuss the results and how they can be interpreted in perspective of previous studies and of the working hypotheses. The findings and their implications should be discussed in the broadest context possible and limitations of the work highlighted. Future research directions may also be mentioned. This section may be combined with Results.
- Materials and Methods: They should be described with sufficient detail to allow others to replicate and build on published results. New methods and protocols should be described in detail while well-established methods can be briefly described and appropriately cited. Give the name and version of any software used and make clear whether computer code used is available. Include any pre-registration codes.
- Conclusions: This section is not mandatory, but can be added to the manuscript if the discussion is unusually long or complex.
- Line 37-39: Please revise the sentence [Due to the presence of polyphenols, tannins, sterols, fatty acids, polysaccharides, these plants also possessed antibacterial, analgesic, anti-inflammatory, antiviral, and anticancer activities].
- Line 276: The available scientific reports confirm this thesis [26=32]. ???Not understood??? Please delete the sentence and discuss with appropriate references.
- Materials and Methods: Please mention the ‘references’ for the procedure followed for Antioxidant activity (DPPH), Surface tension measurements, hydrolysis and extracts purification, TLC analysis.
- Line 379: ‘The quantitative analysis was partially carried out according to references [24,25]. Please write according to de Combarieu et al. 2002 and Gao et al 2015 [24,25].
- Line 392: The qualitative analysis was carried out following the method of Chen et al. 2011 [23] with modification.
- Please remove the GRID line from Fig. 6.
- Please cross-check the references as per the journal style.
- Authors are requested to check the minor spelling (eg. Tannins) and grammar.
The manuscript may be accepted after minor revision.
Author Response
The reply to the Referee
Once again, I would like to thank for the examination of our paper. I tried to revise our manuscript in accordance with reviewer's comments. I hope that the revised work will satisfy the Reviewer.
Responding to the Review 1
- Please follow the ‘Instruction to authors @ Molecules’, The heading ‘Discussion’ is missing. The authors may mention ‘Results and Discussion’ in section 2 or make ‘Result’ and ‘Discussion’ separately.
Answer: We have decided for the ‘Results and Discussion’ in section 2.
[Research Manuscript Sections
- Introduction: The introduction should briefly place the study in a broad context and highlight why it is important. It should define the purpose of the work and its significance, including specific hypotheses being tested. The current state of the research field should be reviewed carefully and key publications cited. Please highlight controversial and diverging hypotheses when necessary. Finally, briefly mention the main aim of the work and highlight the main conclusions. Keep the introduction comprehensible to scientists working outside the topic of the paper.
- Results: Provide a concise and precise description of the experimental results, their interpretation as well as the experimental conclusions that can be drawn.
- Discussion: Authors should discuss the results and how they can be interpreted in perspective of previous studies and of the working hypotheses. The findings and their implications should be discussed in the broadest context possible and limitations of the work highlighted. Future research directions may also be mentioned. This section may be combined with Results.
- Materials and Methods: They should be described with sufficient detail to allow others to replicate and build on published results. New methods and protocols should be described in detail while well-established methods can be briefly described and appropriately cited. Give the name and version of any software used and make clear whether computer code used is available. Include any pre-registration codes.
- Conclusions: This section is not mandatory, but can be added to the manuscript if the discussion is unusually long or complex.
Answer: We made every effort to ensure that the layout and content of the work were in line with the requirements of the journal.
- Line 37-39: Please revise the sentence [Due to the presence of polyphenols, tannins, sterols, fatty acids, polysaccharides, these plants also possessed antibacterial, analgesic, anti-inflammatory, antiviral, and anticancer activities].
Answer: The sentence has been revised.
- Line 276: The available scientific reports confirm this thesis [26=32]. ???Not understood??? Please delete the sentence and discuss with appropriate references.
Answer: The sentence has been revised. A relevant discussion has been added based on the available data.
- Materials and Methods: Please mention the ‘references’ for the procedure followed for Antioxidant activity (DPPH), Surface tension measurements, hydrolysis and extracts purification, TLC analysis.
Answer: References suitable for mentioned procedures have been added.
- Line 379: ‘The quantitative analysis was partially carried out according to references [24,25]. Please write according to de Combarieu et al. 2002 and Gao et al 2015 [24,25].
Answer: The sentence has been revised.
- Line 392: The qualitative analysis was carried out following the method of Chen et al. 2011 [23] with modification.
Answer: The sentence has been revised.
- Please remove the GRID line from Fig. 6.
Answer: The Figure 6 has been revised.
- Please cross-check the references as per the journal style.
Answer: References as per the journal style have been changed. We apologize for this oversight on our part, but we were guided by the previously manuscripts.
- Authors are requested to check the minor spelling (eg. Tannins) and grammar.
Answer: Proposed manuscript has been checked by the professional translator.
